# The Microbial Degradation of Natural and Anthropogenic Phosphonates

**DOI:** 10.3390/molecules28196863

**Published:** 2023-09-29

**Authors:** Francesca Ruffolo, Tamara Dinhof, Leanne Murray, Erika Zangelmi, Jason P. Chin, Katharina Pallitsch, Alessio Peracchi

**Affiliations:** 1Department of Chemistry, Life Sciences and Environmental Sustainability, University of Parma, I-43124 Parma, Italyerika.zangelmi@unipr.it (E.Z.); 2Institute of Organic Chemistry, Faculty of Chemistry, University of Vienna, A-1090 Vienna, Austria; tamara.dinhof@univie.ac.at; 3Vienna Doctoral School in Chemistry (DoSChem), University of Vienna, A-1090 Vienna, Austria; 4School of Biological Sciences and Institute for Global Food Security, Queen’s University Belfast, 19 Chlorine Gardens, Belfast BT9 5DL, UK

**Keywords:** phosphonate catabolism, phosphonate transporters, phosphonatase, oxidative C–P bond cleavage, C–P lyase, environmental distribution of phosphonate degradation, degradation of anthropogenic phosphonates, glyphosate

## Abstract

Phosphonates are compounds containing a direct carbon–phosphorus (C–P) bond, which is particularly resistant to chemical and enzymatic degradation. They are environmentally ubiquitous: some of them are produced by microorganisms and invertebrates, whereas others derive from anthropogenic activities. Because of their chemical stability and potential toxicity, man-made phosphonates pose pollution problems, and many studies have tried to identify biocompatible systems for their elimination. On the other hand, phosphonates are a resource for microorganisms living in environments where the availability of phosphate is limited; thus, bacteria in particular have evolved systems to uptake and catabolize phosphonates. Such systems can be either selective for a narrow subset of compounds or show a broader specificity. The role, distribution, and evolution of microbial genes and enzymes dedicated to phosphonate degradation, as well as their regulation, have been the subjects of substantial studies. At least three enzyme systems have been identified so far, schematically distinguished based on the mechanism by which the C–P bond is ultimately cleaved—i.e., through either a hydrolytic, radical, or oxidative reaction. This review summarizes our current understanding of the molecular systems and pathways that serve to catabolize phosphonates, as well as the regulatory mechanisms that govern their activity.

## 1. Introduction

Organophosphonates (henceforth termed phosphonates for simplicity) are compounds containing a direct C–P bond instead of the more usual C–O–P ester linkage [1]. Despite a similar dissociation energy of the C–P bond as compared to the O–P bond, the cleavage of the former requires a much higher activation energy [2]. Hence, phosphonates are characterized by a high chemical and thermal stability, in addition to the fact that the C–P bond is resistant to the action of phosphatases. Phosphonates regularly occur in the environment, coming from two main sources: either biological (natural phosphonates) or anthropogenic (man-made phosphonates).

The most common natural phosphonates are shown in Figure 1. They are synthesized by a large variety of microorganisms (including protozoa, archaea, bacteria) as well as by coelenterates, mollusks, and possibly plants and insects [3,4,5,6,7]. Natural phosphonates are particularly abundant and biologically important in freshwater and marine ecosystems [8]. In certain marine environments, for example, phosphonates can represent up to 10% of the total dissolved organic phosphorus (DOP) pool [9,10], which becomes 25% in high-molecular-weight DOP [9]. Phosphonates are also found in soil [11,12], although their concentration is, in general, lower than in aquatic environments.

Anthropogenic phosphonates in the environment derive primarily from large-scale human activities, such as agricultural treatments (e.g., pesticides, insecticides) and industrial processes [13]. Because of the wide use of these compounds and the extremely inert nature of the C–P bond, anthropogenic phosphonates tend to accumulate in the environment and represent a major source of both land and water pollution. Some examples of anthropogenic phosphonates are shown in Figure 2.

While phosphorus is an essential nutrient for living organisms, its simplest bioavailable form (inorganic phosphate) is found in low concentrations in various environments, often limiting microbial growth. To cope with the scarcity of inorganic phosphate, many microorganisms have developed peculiar strategies, one of which is scavenging the phosphorus from environmentally available phosphonates [14,15,16]. Several different types of microorganisms are able to degrade phosphonates, including fungi [17,18] and possibly Archaea [19], but by far the most abundant information has been collected on bacteria. In bacteria, the study is facilitated by the tendency of genes involved in the same metabolism to be grouped in operons or clusters [20].

So far, all the metabolic systems identified as degrading the C–P bond can be subdivided based on the mechanism by which the bond is ultimately cleaved. Accordingly, we schematically distinguish the hydrolytic, oxidative, and radical pathways. The hydrolytic and oxidative pathways are generally specific for one or a few substrates, whereas the radical mechanism of cleavage is typical of the C–P lyase complex, which has a broad substrate scope.

This paper provides an updated overview on the biochemical and regulatory strategies that microbes adopt to degrade phosphonates, as well as the abundance and distribution of phosphonate degradative pathways in natural environments. Our focus is primarily on the catabolism of naturally occurring molecules, but we also provide information on the degradation of anthropogenic compounds. While some phosphonate catabolic pathways have been individually reviewed in recent years [21,22,23], there are few reviews (and no very recent ones) that offer a global perspective on the topic, analogous to the one provided here [1,3,14,24,25]. Among other things, we report on recently described phosphonate transporters, illustrate novel accessory enzymes that expand the substrate range of otherwise substrate-specific pathways, and analyze the output of recent metagenomic studies on the abundance of phosphonate degradative systems. Finally, we propose future avenues of research in the field.

## 2. Bacterial Phosphonate Transporters

An early step in the microbial process of scavenging the phosphorus from phosphonates is the uptake of the latter from the environment, and since biological membranes are generally impermeable to charged substances, the import of phosphonates implies the involvement of transporters. Indeed, the bacterial gene clusters dedicated to phosphonate degradation very often include genes encoding transporter proteins. While direct experimental data on these transporters are quite limited, it is evident that different types of bacterial phosphonate transporters exist, arguably distinct on the basis of structure, specificity, and regulation.

Most of the transporters seem to belong to two main classes: ABC transporters and symporters/antiporters.

### 2.1. ABC Transporters

ATP binding cassette (ABC) transporters are multiprotein complexes that encompass one (or more) membrane-associated ATPase subunits and exploit ATP hydrolysis to drive the translocation of substrates across membranes (in our case, to import phosphonates), potentially against their concentration gradient. A typical ABC transporter consists of four components: a substrate-binding protein (SBP), two transmembrane domains, and an ATPase domain [26]. The SBP binds the phosphonate with high affinity and delivers it to the transmembrane domains, which transport it across the membrane using the energy from ATP hydrolysis [26]. SBPs are associated not only with bacterial ABC importers but also with a family of symporters/antiporters called tripartite ATP-independent periplasmic (TRAP) transporters [27].

There are a few known or putative ABC transporters found in bacterial gene clusters dedicated to phosphonate degradation [28]. The best studied is arguably the PhnCDE transporter (Figure 3), whose genes in many bacteria are associated with the cluster for the broad substrate specificity C–P lyase complex [21]. For example, as detailed in Section 5 below, the genes for phosphonate metabolism in *Escherichia coli* are encompassed in the *phn* operon, whose first three genes are *phnC*, *phnD*, and *phnE.* They encode, respectively, an ATP-binding subunit [29], a periplasmic substrate binding protein, and a phosphonate permease (Figure 3a). The function of *phnE* is cryptic in several *E. coli* strains (such as K-12), which are unable to grow on phosphonates [30]. The schematic mechanism through which the PhnCDE transporter is assumed to work is depicted in Figure 3.

The properties of PhnD from *E. coli* have been studied at the structural, functional, and applicative levels [29,33,34]. It seems rather specific for AEP (see Figure 1), with a K_d_ = 5 nM [33], keeping with the notion that this phosphonate is particularly abundant in nature [5,9,35,36]. By comparison, methylphosphonate (MePn, Figure 1; another common natural phosphonate) showed a K_d_ = 1 µM, whereas the anthropogenic aminomethylphosphonate (AMPA, Figure 2) showed a K_d_ = 5 µM [33], despite its obvious structural resemblance to AEP. The binding of AEP to PhnD brings about a substantial structural change in the protein [29] which is believed to be important for the functioning of the transporter (Figure 3).

Another ABC-type transporter (very distinct from PhnCDE) that has been studied to some extent is PhnSTUV from *Salmonella typhimurium LT2*, whose function in AEP uptake was inferred by Jiang and coworkers through complementation experiments [37,38]. The three-dimensional structure of the substrate-binding protein PhnS is known (PDB 4R6Y), although it has never been described in the primary literature.

The expression of the PhnCDE transporter is sensitive to phosphate levels [30], but this is not true for all ABC transporters for phosphonates. For example, the AepXVW transporter of the marine bacterium *Stappia stellulata* DSM5886 (belonging to the *Hyphomicrobiales*) was highly expressed during both phosphate-sensitive and phosphate-insensitive growth on AEP [39]. The substrate binding protein AepX showed an affinity for AEP comparable to that of PhnD (K_d_ = 23 nM) but a much higher specificity (the K_d_ values for MePn and AMPA were 3.4 mM and 4.4 mM, respectively), consistent with the observation that *aepXVW* genes are commonly associated with either *phnWX* or *phnWAY* clusters, which encode AEP-specific degradation systems (Section 3.1) [39]. Homologs of *aepXVW* are expressed constitutively in diverse marine heterotrophs and are ubiquitously distributed in mesopelagic and epipelagic waters. In the global ocean, *aepX* was heavily transcribed (~100-fold more than *phnD*) independently of phosphate and nitrogen concentrations [39].

### 2.2. Symporters/Antiporters (Facilitators)

Antiporters (which move two substrates across a membrane in opposite directions) and symporters (which transport two substrates in the same direction) do not require ATP hydrolysis but, in general, utilize some ion gradient as a secondary energy source.

Some of these transporters involved specifically in phosphonate uptake have been characterized. In a recent study, Murphy et al. found that the facilitator-type AEP permease (AepP) was essential for the phosphate-insensitive growth of *Pseudomonas putida* BIRD-1 on AEP. The expression of the *aepP* gene occurred both in the presence and in the absence of phosphate, and it significantly increased during growth on AEP as a sole N source (phosphate-insensitive) [39]. A similar gene was identified in the genome of *Sinorhizobium meliloti* 1021 [39]. Indeed, it had been noted before that in *S. meliloti* 1021, the same operon encoding the AEP-specific PhnWAY pathway (see Section 3.1 below) also encompasses the genes for a putative sodium-dependent symporter (NCBI WP_010975820) as well as an ABC transporter [40]. The compresence of this Na^+^ symporter with the genes for an AEP-specific degradation pathway is retained in a number of related species, strongly suggesting that this transporter is involved in AEP uptake. Notably, the Pho regulon is not essential for AEP transport in *S. meliloti* 1021 [41].

Wicke et al. [42] identified another facilitator involved in phosphonate transport by assessing the adaptation of the Gram-positive *Bacillus subtilis* to the herbicide glyphosate (Figure 2). Through analyses at the whole genome level, they found that *B. subtilis* develops resistance to glyphosate primarily by inactivating *gltT*, which encodes a high-affinity glutamate/aspartate symporter. Metabolome analyses confirmed that GltT is the major entryway of glyphosate into *B. subtilis*. The same transporter also appeared to be involved in the uptake of the herbicide glufosinate (a phosphinate, rather than a phosphonate compound). Complementation experiments indicated that the ectopic expression of GltP, the GltT homolog of *E. coli*, restored the glyphosate sensitivity of a *B. subtilis gltT* mutant, suggesting that GltP could also be involved in glyphosate uptake in *E. coli* [42].

## 3. Hydrolytic Pathways for the Catabolism of Phosphonates

In compounds such as PnPyr or phosphonoacetate (PnAc; Figure 1), the reactivity of the C–P bond is increased by the presence of a neighboring carbonyl or carboxyl group, making its cleavage through a hydrolytic mechanism possible. Analogously, the corresponding aminophosphonates (PnAla or AEP) can be activated by converting the amino group to a carbonyl or carboxyl functionality. In the hydrolytic pathways of aminophosphonate degradation, such an initial activation precedes the cleavage of the C–P bond and the assimilation of phosphate. This metabolic activation strategy is common in microbial biodegradation pathways and is observed, in addition to phosphonates, with aromatic hydrocarbons and many other compounds. Among the phosphonate hydrolytic pathways, those targeting AEP are unsurprisingly the most abundant ones, since AEP is the most widespread natural phosphonate [5,9,36].

### 3.1. Degradation of AEP through the PhnWX and PhnWAY Pathways

The two best-known routes for AEP catabolism are sometimes called the PhnWX and PhnWAY pathways (from the gene symbols of the involved enzymes). The two pathways are partially overlapping in that they both start with the transamination of AEP, catalyzed by the enzyme PhnW (2-aminoethylphosphonate:pyruvate aminotransferase), to yield PAA and alanine (Figure 4a). PhnW uses pyridoxal phosphate (PLP) as the cofactor, and its reaction involves the formation of a Schiff base between PLP and the β-carbonyl group of AEP. Subsequent steps lead to the release of PAA with the concomitant formation of the aminated form of the cofactor (pyridoxamine phosphate). In the second half of the reaction, pyruvate binds to the active site, forming a Schiff base with the pyridoxamine. The Schiff base is then hydrolyzed, and alanine is produced [38]. Structural analysis of PhnW from both *Salmonella typhimurium* [43] and *Pseudomonas aeruginosa* PAO1 [44] revealed that the enzyme is catalytically active as a homodimer, and each monomer has a tertiary structure common to the majority of transaminases [43,44]. It involves a large domain for PLP binding and a smaller domain that switches from an open to a closed conformation upon substrate binding. The active site is located between the two domains [43,44].

The PhnWX pathway was initially described in *Bacillus cereus*, an organism that could use AEP as the sole P source [47,48]. In this pathway, subsequent to the PhnW reaction, the Mg^2+^-dependent enzyme phosphonoacetaldeyde hydrolase (PhnX) cleaves PAA into acetaldehyde and phosphate (Figure 4a). A number of studies have led to a better understanding of the mechanism and structure of PhnX [37,48,49,50,51]. In its reaction, an unprotonated catalytic lysine generates a Schiff base with PAA. The activated C–P bond is then hydrolytically cleaved through a highly conserved active-site Asp residue, thus releasing the final products [49,50,52].

An analysis of the *S. meliloti* genome sequence revealed the absence of a *phnX* homolog, yet the organism was able to grow using AEP as the sole phosphorus source. In this case, the AEP degradation took place alternatively through the PhnWAY pathway, which could also be the main route for PnAc production in nature [40,53,54]. Indeed, in this pathway, PAA is oxidized to PnAc by the NAD-dependent PAA dehydrogenase (PhnY) (Figure 4a). Finally, phosphonoacetate hydrolase (PhnA) catalyzes the Zn^2+^-dependent hydrolytic cleavage of PnAc, yielding inorganic phosphate [40,55].

PhnA had been discovered earlier than the whole pathway. *Pseudomonas fluorescens* 23F was the first isolated environmental microbe capable of using PnAc as a C and P source, converting it to phosphate and acetate [56,57]. Further studies led to the purification and characterization of *P. fluorescens* 23F PhnA, which was able to degrade only PnAc and, to a lesser extent, 2-phosphonopropionate [53,58]. Homologs of the *phnA* gene were detected in soil isolates of several geographical regions as well as in coral-living microbes, without enrichment on PnAc. The latter was considered a synthetic phosphonate compound (used as an antiviral medication) [59] However, the ubiquitous presence of *phnA* genes in isolates from distinct geographical regions as well as the high specificity of the enzyme toward PnAc indicated that PnAc can have a biogenic origin [59,60].

The PhnWX and PhnWAY pathways do not co-occur among marine genomes, likely due to their superimposable functions [36]. Moreover, it has been suggested that the two pathways are taxonomically separated among proteobacteria; while the *phnWX* pathway is particularly frequent in *Vibrionales*, *Oceanospirillales*, and *Alteromonadales* (all *Gammaproteobacteria*), the *phnWAY* pathway has been found in *Alphaproteobacteria*, specifically in *Rhodobacterales* species [36].

### 3.2. Ancillary Enzymes Can Expand the Usefulness of AEP Degradation Pathways

While the PhnWX and PhnWAY pathways are broadly diffused, they are also apparently limited in versatility, as they only serve to process AEP (or, in the case of the PhnWAY pathway, AEP and PnAc). However, as in other branches of catabolism (e.g., [61,62]), the usefulness of the pathways might be improved by ancillary enzymes that convey other types of substrates into the pipeline. A case in point is represented by the identification of a PLP-dependent lyase, termed PbfA, whose gene is recurrently found adjacent to the *phnWX* gene cluster in many *Gammaproteobacteria* and to the *phnWAY* cluster in many *Alphaproteobacteria*. This enzyme was shown to catalyze an elimination reaction on its specific phosphonate substrate *(R)*-HAEP (Figure 1), yielding PAA and ammonia and thereby expanding the substrate scope of the pathway [45] (Figure 4a).

On a similar note, a recent study focused on three groups of FAD-dependent oxidoreductases, named PbfB, PbfC, and PbfD, whose genes were often found associated with gene clusters of the *phnWX* and *phnWAY* types and in some cases to just *phnX* [46]. Even though these three groups of enzymes were quite different sequence-wise, all of them appeared capable of converting M_1_AEP into PAA through a redox reaction, leading to methylamine release (Figure 4a). However, they showed different functional properties regarding their preference to employ (or not) molecular oxygen as the electron acceptor and the ability to also use AEP as their substrate. One functionally characterized PbfD-type enzyme was able to oxidize M_1_AEP and AEP with similar efficiencies, producing PAA in both cases. Such PbfD may effectually replace the PhnW activity in those microorganisms where the *phnW* gene is absent [46].

### 3.3. Hydrolytic Degradation of Phosphonoalanine (PnAla)

A distinct hydrolytic pathway allows for the utilization of PnAla and PnPyr (Figure 1). PnAla is a component of phosphonolipids, and it is particularly abundant among marine microbes and invertebrates [25,63]. This pathway was first identified in the environmental isolate *Burkholderia cepacia* Pal6 and involves two enzymatic steps (Figure 4b) [64]. As in the case of AEP mineralization, the first step is a PLP-dependent transamination reaction catalyzed by the transaminase PalB, which converts PnAla into PnPyr. The latter is then hydrolyzed into pyruvate and phosphate by the Co^2+^-dependent phosphonopyruvate hydrolase (PalA) [65,66]. PalA was first purified from *Variovorax sp*. Pal2 and showed sequence similarities with several phosphoenolpyruvate mutase superfamily members; the enzyme was very specific for PnPyr [65].

Although the three substrate-specific phosphonohydrolases described above (PhnX, PhnA, and PalA) are all metal ion-dependent and catalyze similar enzymatic reactions, these enzymes are not evolutionarily related. In fact, unlike PalA, PhnX belongs to the haloacid dehalogenase (HAD) family, which uses an Asp as the catalytic residue contained in the active site. Instead, PhnA is part of the alkaline phosphatase superfamily of Zn^2+^-dependent hydrolases and is likely to use a Thr residue as the nucleophile to attack PnAc [16,23,32]. Specifically, the X-ray structure of the *P. fluorescens* 23F PhnA revealed its high similarity with the metalloenzyme nucleotide pyrophosphatase/diesterase [50,55,65]. Nonetheless, the PhnA chemical function is new among the alkaline phosphatase superfamily, and it differs in the stabilization of the leaving group, which, in the case of PnAc, is an aci-carboxylate dianion. This intermediate, indeed, should be stabilized by two Lys residues instead of one of the zinc ions, as in the case of the phosphate esters compounds [55].

## 4. Oxidative Pathways for Phosphonate Catabolism

The oxidative C–P bond cleavage pathway was discovered most recently. In 2010, during screenings of metagenomic marine DNA libraries, Martinez et al. identified the first set of genes involved in this process, *HF130*PhnY* and *HF130*PhnZ [8]. The encoded enzymes [36] and homologs thereof are commonly found in marine bacteria and catalyze a two-step reaction sequence. The α-ketoglutarate/non-heme Fe^II^-dependent dioxygenase PhnY* performs a stereospecific hydroxylation of the α-carbon of AEP (Figure 5a) during which O_2_ and the co-substrate, α-ketoglutarate, bind to the active site bound Fe^II^. Then, molecular oxygen is reduced to superoxide, followed by the oxidative decarboxylation of α-ketoglutarate to form succinate. This leads to the highly reactive Fe^IV^ = O species, which is able to abstract the α-hydrogen of AEP, upon which an a Fe^III^-OH-species and a radical in α-position to the phosphorus of the substrate are formed. The hydroxy group is immediately transferred to the substrate in a radical reaction, which provides the enantiopure intermediate (*R*)-HAEP, which is the substrate for PhnZ [67,68]. This enzyme is nowadays classified as a member of the mixed valent (Fe^II^/Fe^III^) diiron-dependent oxygenases (MVDOs) [69] and thus belongs to the histidine-aspartate (HD) superfamily of metalloproteins [70]. Another prominent example of this family is the mammalian enzyme myo-inositol oxygenase (MIOX), which forms D-glucuronic acid by the oxidative cleavage of the C1–C6 bond of myo-inositol [71].

The mechanism of C–P bond cleavage by PhnZ is still not fully elucidated, but several computational, structural, and biochemical studies led to the following assumptions discussed in detail in a very recent review by Pallitsch and Zechel [23]. While PhnY* is highly substrate-specific, PhnZ shows some substrate tolerance but high specificity towards (*R*)-configured α-hydroxyphosphonates. In brief: PhnZ undergoes three main stages: first, a conformational change in the active site loops is promoted during substrate-binding; then, the activation of dioxygen is followed by α-H abstraction of the substrate and, finally, by C–P bond cleavage to form inorganic phosphate and glycine.

Soon, PhnY*/PhnZ homologs with different substrate specificities were discovered, and today, four versions of this pathway are known. Gama et. al. showed that the microorganism *Gimesia maris* DSM8797 encodes the enzyme pair *Gm*PhnY* and *Gm*PhnZ. While *Gm*PhnY* is specific for MePn, the corresponding PhnZ-variant is able to cleave both (*R*)-HAEP and hydroxymethylphosphonate (HMP) [72] (Figure 5a,b). This is the only known pathway for MePn degradation apart from the C–P lyase (discussed in Section 5). However, here, the final products of MePn degradation are inorganic phosphate and formate, generated via the intermediate HMP. Intriguingly, and in contrast to C–P lyase, the byproduct, formate, can also be used as a carbon source during purine biosynthesis by the host bacterium [73].

In *Leisingera caerulea*, another very similar set of enzymes was discovered by Rajakovich et al. in 2019 [74,75]. The respective two-gene operon encodes the enzymes TmpA and TmpB. The highly substrate-specific dioxygenase TmpA performs the stereospecific hydroxylation of M_3_AEP to (*R*)-M_3_HAEP, followed by C–P bond cleavage to inorganic phosphate and glycine betaine by TmpB (Figure 5c).

*phnY*/phnZ*-like genes are, however, not uniquely found in bacteria. Recently, a fourth version of this pathway was discovered in *Fonsecaea multimorphosa*, a fungal pathogen [76]. In this special case, PhnY* and PhnZ are fused to the single bifunctional enzyme called PhoF. Besides AEP, APP can also serve as a substrate for PhoF (Figure 5d), making its substrate-scope unique among all known oxidative pathways.

## 5. The C–P–Lyase Complex

Of all C–P bond-cleaving enzymes identified to date, the carbon–phosphorus lyase (C–P lyase) complex has garnered prominent attention in the fields of biogeochemistry and environmental science [77,78,79,80]. By comparison to the substrate-specific enzymes discussed above, C–P lyase displays a high degree of promiscuity with a remarkable substrate range [25,81]. This activity was first described by Wackett et al. [82], who documented the consumption of structurally diverse alkyl, aminoalkyl, and arylphosphonates in various bacterial isolates. Subsequent attempts to purify the C–P lyase and study, in vitro, its activity were unsuccessful, owing to the poor solubility of the central C–P cleavage proteins. In general, the study of the C–P lyase has been hindered by the intricacy of the enzyme system’s various interacting protein domains and enzymes coupled with the need for anaerobic reaction conditions. In recent years, however, significant advances have been made in understanding the major structural features and the catalytic mechanism of the multienzyme complex [83,84].

### 5.1. Organization of the C–P Lyase Complex

Intensively studied and best characterized in *E. coli*, the subunits of the C–P lyase complex are encoded within the 14-gene *phnCDEFGHIJLKMNOP* operon (Figure 6), first elucidated by Metcalf et al. [85]. However, variant operon structures are known to exist in taxonomically diverse bacterial species and some halophilic Archaea (Figure 6) [77,86,87]. The expression of the co-transcribed genes is upregulated in response to P-limitation [88]. To date, Pho regulation (see Section 7) is the only known mechanism of the transcriptional control of C–P lyase activity. Despite this, a study by Karl et al. [89] demonstrated the downregulation of C–P lyase activity but not total abolishment within marine surface water cultures supplemented with MePn (Figure 1) and phosphate, which suggests that alternative and possibly dual regulation mechanisms of the C–P lyase exist.

Among the 14 proteins encoded by the C–P lyase complex in *E. coli*, PhnCDE functions as an ABC transport system, as described in Section 2. This transport system has been found to be negatively regulated by PhnF, which functions as an operon repressor [91]. Of the ten proteins encoded by *phnGHIJKLMNOP*, a subgroup, specifically PhnGHIJKLM, is critical for enzymatically cleaving the C–P bond [92]. In recent years, the function and crystal structure of the 220 kDa catalytic core complex (PhnGHIJ) and the proteins involved in exposing the active site of the core complex (PhnKL) have been resolved [83,84,93].

### 5.2. Reaction Mechanism by the ‘Central’ C–P Lyase Subunits

The consumption of phosphonates by the C–P lyase pathway in *E. coli* occurs in an ATP-dependent manner, converting phosphonates into 5-ribosyl-α-1-diphosphate (PRPP), a central metabolite in intermediary metabolism (Figure 7) [83].

The initial step of the C–P lyase reaction is initiated by the nucleotide phosphorylase PhnI, which catalyzes the displacement of adenine from ATP, forming D-ribose-5-triphosphate (RTP; Figure 7). In the presence of PhnGHL, the phosphonate substrate (e.g., MePn) is coupled with RTP, forming the triphosphate ester α-D-ribose-1-methylphosphonate-5-triphosphate (RPnTP). The hydrolysis of the triphosphate moiety by the phosphohydrolase PhnM yields pyrophosphate and α-D-ribose-1-methylphosphonate-5-phosphate (PRPn) [94]. The latter serves as the substrate for PhnJ, in which the C–P bond is cleaved via a strictly anaerobic S-adenosyl methionine (SAM)-dependent glycine-radical reaction mechanism to yield the alkyl group (in the case of MePn, CH_4_) and α-D-ribose-1,2-cyclic-phosphate-5-phosphate (PRcP) [92].

Essential for C–P lyase catalysis via PhnJ are PhnK and PhnL, two non-transport ABC proteins with ATPase activity, which bind simultaneously to the core complex, where the hydrolysis of ATP exposes the active site between PhnI and PhnJ upon the opening of the catalytic core [84]. This activity is essential for *E. coli* growth on phosphonates. Forming the end stages of the reaction and the incorporation of the phosphonate-derived phosphorus atom into the intermediary metabolism in *E. coli*, the cyclic phosphate PRcP is hydrolyzed by the phosphodiesterase PhnP, yielding RbP [95]. The phosphorylation of RbP by the ribose bisphosphokinase PhnN yields the central metabolite PRPP [96].

### 5.3. Role of Nonessential Subunits in the C–P Lyase Function

The PhnO protein is an N-acetyltransferase which is non-essential for the catabolism of phosphonates with the amino group at carbon 2 or higher such as AEP. The role of PhnO is the detoxification (via acetylation) of 1-aminoalkylphosphonates such as AMPA (Figure 2), a bacteriocidal breakdown product of glyphosate, before utilization as a source of phosphate [97,98]. Thus, PhnO performs an important although optional accessory function depending on the organism and on the availability of substrates. For example, *S. stutzeri* is deficient in PhnO (Figure 6) and lacks the ability to grow on AMPA as a source of phosphate [77]. Similarly, PhnP and PhnN are replaced in some organisms harboring variant C–P lyase operons. For example, the marine cyanobacterium *T. erythraeum* is capable of growth on MePn and AEP as sources of phosphate, despite lacking the PhnNOP proteins [99]. Additionally, within C–P lyase operons lacking *phnP*, the *rcsF* gene (encoding a member of the 2Hphosphodiesterase superfamily) has been noted to be uniquely associated with *phn* operons. A survey of the SEED database using PhnM as a query showed that, of a total of 54 bacterial operons, 27 of these co-occurred with PhnP. However, the *rcsF* gene was present within the 16 operons lacking *phnP*. This family of proteins can hydrolyze ribosyl-1′, 2′-cyclic phosphates and could potentially process the cyclic phosphodiester intermediate PRcP [95]. While best studied in bacteria, the consumption of phosphonates may be extended to archaeal organisms—for example, *H. walsbyi* [19] (Figure 6) and *Natronomonas moolapensis* [77] possess the core genes *phnGHIJKLM*, which are required for C–P cleavage. Analogously to the cyanobacterium *T. erythraeum* (Figure 6), both archaeal strains lack the accessory genes *phnNOP* and the regulatory protein PhnF [77]; however, there are no published data for archaeal phosphonates consumption.

The unique capacity of the C–P lyase complex to metabolize various phosphonates provides a distinct advantage to microbial species which can hence exploit both biogenic and anthropogenic phosphonates as a source of phosphate. This ability is particularly important in oceanic ecosystems which are either chronically or transiently depleted of phosphate [88,100]. The utilization of phosphonates via C–P lyase has a significant role in both phosphorus and carbon biogeochemistry coupled with aerobic CH_4_ production [89,101,102]. Attracting considerable scientific interest is the role of C–P lyase in the microbial mineralization of the simplest C–P bond-bearing compound, MePn, which is proposedly responsible for supersaturating CH_4_ concentrations in oceanic surface waters [103]. Until recently, C–P lyase was the only known biological mechanism to metabolize MePn as a source of phosphate. However, a significant revelation in terms of C–P biochemistry emerged when a novel oxidative mechanism of MePn catabolism was unveiled in the organism *G. maris* DSM8797 by Gama et al. [72] [Section 4].

## 6. Genomic and Environmental Distribution of the Known Phosphonate Degradation Pathways

The advancements in our understanding of phosphonate degradation pathways naturally raised questions about the importance of these processes for microbial communities and biogeochemical cycling. A search of bacterial genomes in the Joint Genome Institute’s Integrated Microbial Genomes and Microbiomes database found more genomes containing the *phnJ* gene than any other (4139; *phnX =* 3452 genomes; *phnZ =* 2688 genomes; *phnA =* 1524 genomes; *palA =* 201 genomes). Notably, the majority of these homologs belonged to Classes in the Phylum *Pseudomonadota*, with a combined total of 99.0% of *palA*, 85.0% of *phnJ*, 81.6% of *phnZ*, 80.6% of *phnA*, and 71.4% of *phnX* homologs belonging to the *Alpha*, *Beta*, and *Gammaproteobacteria*. The *Pseudomonadota* are highly diverse and abundant across ecosystems [104], suggesting that phosphonate metabolism is likely to be similarly widely distributed. The abundance of phosphonate catabolism genes in non-*Pseudomonadota* was highest in the *Bacilli* (14.7% of *phnX* genes) and the *Actinomycetia* (4.3% of *phnA* homologs) [36].

These trends are consistent with a previous study of genomes in the NCBI GenBank database [3] and with isolation experiments which predominantly recovered phosphonate-consuming *Pseudomonadota* such as *Pseudomonas*, *Roseovarius*, and *Sulfitobacter* species [16,59]. Despite this, some other taxa which are highly adapted to low phosphorus levels may also possess phosphonate catabolism genes, such as *Trichodesmium* spp. in the Class *Cyanophyceae* [105]. However, databases of sequenced isolates are known to be biased towards medically and industrially important organisms [106], and while growth experiments do suggest that bacteria from a wide range of environments can consume phosphonates, these studies fail to culture representatives of many taxa [107,108]. The development of metagenomic sequencing has allowed for the quantification of phosphonate catabolism genes from a much wider range of taxa within diverse ecosystems to better understand their environmental role.

### 6.1. Studies in Marine Systems

Large-scale marine studies have provided metagenomes from water samples across the globe, accompanied by metadata such as inorganic nutrient levels, which can be used to determine the environmental factors which drive gene abundance. Studies of these datasets generally compare the abundance of a gene of interest to markers of cell abundance, yielding estimates of the proportion of the microbial community with that gene. While these studies have shown variations in precise abundance estimates, likely due to differences in sampling and data processing approaches, clear trends are apparent (Table 1).

The Global Ocean Sampling (GOS) expedition collected surface water samples from a transect across the North Atlantic to the South Pacific oceans [109]. Two analyses of the ensuing metagenomic data found that *phnA* was the most abundant phosphonate catabolism gene, followed by *phnJ* and then *phnX* (9%, 8%, and 1% of expected bacteria, respectively, for Martinez et al. [8], and 11.2%, 5.1%, and 1.8%, respectively, for Villarreal-Chiu et al. [3]). A qualitatively consistent result was obtained from an analysis of global surface seawater samples in the Tara Oceans dataset [110] by Murphy et al. [39], who found that the median abundance of *phnA* was slightly under 10% of expected bacterial cells, with median abundances of *phnJ* and *phnX* less than 1% and 0.1%, respectively. However, another analysis of the same data by Lockwood et al. [36] found the average *phnJ* abundances to be over 10% of expected cells, followed by *phnA* and then *phnX* (approximately 10% and 2% of cells, respectively), emphasizing the impact that different data processing approaches can have on metagenomic analysis (Table 1). The *palA* gene was only examined by Villarreal-Chiu et al. and Lockwood et al., who found that it was not abundant in surface waters (0.1% and <5% of expected genomes, respectively).

Significantly different abundances have also been estimated for the *phnZ* gene. Martinez et al. [8] and Villarreal-Chiu et al. [3] estimated that approximately 7% and 9.4%, respectively, of expected cells in the GOS data possessed *phnZ*, making it only slightly less abundant than *phnA*. By contrast, Lockwood et al. [36] found that ~20% of expected bacteria in the surface samples possessed it, making it the most abundant enzyme-coding gene in their analysis by a factor of two. However, Murphy et al. [39] noted that after AEP catabolism by PhnZ was discovered, homologs of *phnZ* were found, which were specific for different substrates (MePn in the case of the *G. maris*-like *phnZ*, or M_3_AEP in the case of *tmpB*; discussed in Section 4). Murphy and colleagues differentiated these subtypes and found a median AEP-*phnZ* abundance of <0.01% of bacteria in surface waters, making it the least abundant enzyme-coding gene in their dataset. They further estimated that the *G. maris* and *tmpB phnZ*-like genes were present in <1% of expected bacteria. Even collectively, the total abundance of *phnZ*-like genes in this study was significantly lower than in other studies, leaving estimates for *phnZ* abundance ranging from one of the least abundant to the most abundant phosphonate catalytic pathway in surface waters. This makes it difficult to form a consensus on the abundance of *phnZ*-like pathways in surface oceans.

The pathways for phosphonate degradation were initially thought to help marine bacteria survive in surface waters by allowing for the consumption of phosphonates when orthophosphate was scarce [111]. This would suggest that deeper waters, which tend to be more orthophosphate-rich [112], would have lower abundances of these genes. However, some strains of marine bacteria have been shown to consume phosphonates regardless of phosphate levels to acquire nitrogen, which can also be limiting in marine environments [39,113]. The Tara Oceans data do include samples from the mesopelagic, the layer of water approximately 200 to 1000 m deep which receives little to no sunlight [110]. Using these samples, Murphy et al. [39] and Lockwood et al. [36] observed either no change or an increase in the abundance of all of the enzyme-coding genes, with *phnA* and *phnX* being more abundant than *phnJ*, and the abundance of *phnZ* remaining inconsistent. This suggests a continued or increased consumption of phosphonates as nutrients sink down the water column, rather than a reduction.

NMR studies of the change in the seawater phosphonate concentration with depth have provided inconsistent results [9,114,115], but it was shown using ^33^P-labeled orthophosphate that some marine microbes synthesize reduced P compounds, including phosphonates, in greater quantities at a 150 m depth than a 40 m depth [116]. This suggests that marine phosphonate abundance is controlled by a complex interaction between the microbial community members capable of synthesis and/or degradation, which will likely vary according to the local depth and nutrient abundances. For example, in the Mediterranean Sea, which is extremely phosphate-limited, *phnJ* is the most abundant catabolic gene [36], and so presumably, an increase in broad substrate-range phosphonate catabolism leads to a more wide-ranging reduction in phosphonate abundance than if AEP-specific metabolism was predominant.

### 6.2. Studies in Freshwater and Soil Systems

Compared to the global-scale analyses of metagenomic datasets from marine environments, the analysis of phosphonate catabolic gene abundance in other environments is significantly more limited. A number of studies identified phosphonate metabolism, largely via C–P lyase, as an important phosphorus starvation survival mechanism in freshwater systems [117,118,119], but the abundance of different catabolism genes was not analyzed in detail. Phosphonate metabolism, predominantly via C–P lyase, also appears to be an important survival mechanism in serpentinizing hydrothermal systems [120]. Divergent phosphonate catabolism behavior has been observed in two strains of *Synechococcus* spp. inhabiting a hot spring microbial mat [121], suggesting important roles for phosphonates as both carbon and phosphorus sources. While this demonstrates that phosphonates are important in diverse aquatic habitats, they remain less well characterized than in marine habitats.

Similarly, while many of the bacteria in which phosphonate metabolism was first characterized were terrestrial bacteria, there are few studies of soil phosphonate metabolism. One study of global soil metagenomes observed that cropland and grassland soils had a relatively low functional redundancy of C–P lyase genes, an indication that relatively few taxa of the microbial community possessed the pathway and, therefore, that it was potentially less important for community stability. Conversely, in forest soils, the genes showed similar levels of redundancy to the *phoD* and *phoX* alkaline phosphatases, suggesting a more significant role for C–P lyase in soils which may be more depleted in phosphoesters [122]. However, estimates of the abundance per expected bacterial cells were not produced. Similarly, a study of microbial community resilience in a Chinese grassland soil sample observed that genes categorized as part of “phosphonate and phosphinate metabolism” by the Kyoto Encyclopedia of Genes and Genomes were important for maintaining community stability, although no specific genes were identified [123]. Oliverio and coworkers [124] examined metagenomes from 275 Australian soil samples, observing that genes in the “alkylphosphonate utilization” SEED category were approximately a tenth as abundant as those in the “phosphate metabolism” category, although again, specific genes were not indicated. However, the C–P lyase pathway and *phnX* gene were implicated as important for soils that were low in phosphate, suggesting that it is a major survival mechanism for soil bacteria, similar to some aquatic environments.

The majority of soil phosphonate studies have focused on environmental pollutants such as glyphosate, which is generally degraded via C–P lyase (discussed in Section 8), and so the study of substrate-specific pathways has been significantly neglected in this environment [125]. Better characterized is phosphonate synthesis, with one study suggesting that a diverse range of phosphonates are likely to be produced in soils [4]. It would seem logical that the soil microbiota would subsequently consume these compounds, but a better characterization of what these compounds are, as well as a quantitative study of the abundance of catalytic genes, is needed before this can be confirmed.

## 7. Regulation of Phosphonate Degradation Pathways

As in the case of other metabolic networks, the expression of bacterial genes involved in phosphonate catabolism is under tight control. Such control can vary substantially depending on the organism and on the pathway and can be complex and multi-layered [126,127]. Below, we try to provide some general principles on the phenomenon, while illustrating some well-studied examples of regulation.

### 7.1. Phosphonate Catabolism Is Usually under the Pho Regulon Control

The C–P bond cleavage systems were thought to be regulated by the two-component system *phoBR* [24]. It regulates the expression of genes, which together are part of the so-called Pho regulon, to acquire inorganic phosphate by catabolizing alternative P sources when the cells are under phosphate-starved conditions [128]. In the case of *E. coli* K12, among the genes expressed during phosphate-limiting conditions, there is the C–P lyase enzymatic complex, having the Pho box promoter upstream of the *phnC* gene encoding a phosphonate transporter [22,129] (see Section 2.1). Thus, the C–P lyase is part of the *E. coli* Pho regulon and allows the organism to grow on MePn as the P source [129]. Studies on the Pho^+^ *E. coli* strains and *phoB* and *phoR-phoM E. coli* mutants have, in fact, shown that C–P lyase synthesis is induced only in Pho^+^ bacteria while failing to be expressed in the other mutants [129]. In parallel, growth studies have shown that when in the presence of a growth-limiting concentration of phosphate and MePn, the organism had a lag phase with the concomitant starting of methane production after the phosphate was completely consumed [129]. A similar result was obtained with in vivo experiments on the *Kluyvera ascorbata* C–P lyase: in the presence of limiting concentrations of both phosphate and ethyl phosphonate, the organism started to produce ethane, accompanied by a smaller lag phase, after consuming the phosphate [82]. These results are indicative of the C–P lyase induction under only phosphate-starved conditions, allowing phosphonate compounds to be utilized only as alternative sources of P [82,129].

It has been shown that *Enterobacter aerogenes* possesses the gene operon for the broad-specificity C–P lyase, as well as the *phnWX* cluster, specific for AEP degradation [130]. Like in *E. coli*, the *E. aerogene*s C–P lyase as well as PhnX seem to be under the Pho regulon control [130]. Gene homologs for the PhnWX pathway of *E. aerogenes* were also found in *Salmonella typhimurium* [6]. As in the case of *E. aerogenes* [130], since the plasmid carrying the *S. typhimurium phnWX* operon was unable to allow for the growth of *ΔphoBR E. coli* mutants on AEP as the P source, the *S. typhimurium* PhnWX pathway may also be controlled by its Pho regulon, which acts similarly to that of *E. coli* [37]. Studies on *P. fluorescens* 23F revealed that *phnX* was only expressed under phosphate-starvation, rather than being phosphate-independent and substrate-inducible like *phnA*, as described below [131]. A detailed ‘omics’ study on three different *Pseudomonas* strains revealed remarkably variegated responses to phosphate depletion, suggesting that Pho regulation extended to several proteins (such as putative hydrolases, putative phosphonate transporters, and outer membrane proteins) previously not associated with P starvation [132].

### 7.2. Substrate-Specific Hydrolytic Pathways Expression Can Be Phosphate-Independent and Substrate-Inducible

The assumption that the C–P bond cleavage only occurs when phosphate is depleted has led to the inference that phosphonate degradation is limited to environments where phosphate is the limiting growth factor, such as in many marine environments; thus, phosphonates would be used only as a P source and not as C or N sources, since phosphate release does not allow for the complete mineralization of these compounds [133]. However, phosphonate degradation does actually occur in a phosphate-independent and substrate-inducible manner, and the enzymes involved in the phosphate-insensitive C–P bond cleavage are usually specific for a single phosphonate compound [134]. Studies on the distribution of genes for C–P cleavage in microbial genome sequences, as well as in metagenomic databases, revealed the presence of enzymes for phosphonate degradation even in marine areas where the growth-limiting element is not P but rather N [3,8,39]. In these oligotrophic environments, the phosphate-independent induction of genes for aminophosphonate degradation allows microbes to fully mineralize these compounds to have usable forms of C, N, or P [3,8,39]. Enrichment experiments using AEP as the sole N source and KH_2_PO_4_ as the P source led to the isolation of seven *Alphaproteobacteria* strains [113]. It was shown that all the isolates could mineralize AEP through the PhnWAY pathway since only inorganic phosphate and acetic acid were detected as products, which is consistent with metagenomic studies revealing that *phnA* is the most abundant gene for phosphonate catabolism among marine microbes [113]. Further studies revealed that a number of other *Alphaproteobacteria* possessing *phnWX* and *phnWYA* operons were also able to grow on AEP as the sole N source in the presence of phosphate [39]. In a recent, detailed study [126], Murphy et al. provided evidence that the AEP catabolism in *Pseudomonas putida* BIRD1 is not only under the PhoBR regulation in phosphate-limiting conditions. Specifically, *phnWX* gene expression also occurred under C or N depletion, and it was regulated by the two-component system regulators CbrAB and NtrBC, respectively [126].

The finding that aminophosphonate catabolism can occur in phosphate-rich marine environments increases the importance of these compounds not only in the P cycle but also in those of bioavailable N and C.

Studies on the *P. putida* NG2 and *P. aeruginosa* PAO1 PhnWX pathway for AEP degradation led to opposite conclusions to those obtained from studies on *E. aerogenes* and *S. typhimurium*, proving the presence of phosphate-independent C–P cleavage not only in marine environments but also in soil bacteria. In fact, in the two *Pseudomonas* isolates, the AEP degradation through the PhnWX pathway occurs regardless of the phosphate levels, allowing the cell to use AEP as a C, N, and P source [133,135]. In these organisms, the PhnWX operon was induced only when AEP was present in the medium and did not require phosphate starvation [133,135].

Another study on the degradation of several phosphonate compounds, both natural and synthetic, led to the discovery of environmental microbes capable of utilizing four of the studied phosphonates (AEP, PnAc, PnAla, and phosphonomycin) as C and energy sources with a detectable phosphate release [136].

PhnA, the key enzyme for PnAc utilization, was first found in the soil isolate *P. fluorescens* 23F, as described above. Afterward, other soil isolates from different regions were tested for PhnA activity, and all of them could utilize PnAc as C, P, and energy source, with phosphate release [59]. The expression of the *phnA* gene in the *P. fluorescens* 23F isolate was shown to be substrate-inducible and independent of the presence of phosphate in the medium [53,131]. Hence, the *phnA* gene expression was not under Pho regulation, since it was not influenced by the presence of phosphate. Not only is *phnA* expression phosphate-independent, but it requires the presence of PnAc as well [53,127,131]. Moreover, as the amount of PnAc in the medium increased, there was an increased activity level of PhnA in an *Agromyces fucosus* strain [127]. On the other hand, the same study on this organism has also revealed a possible mechanism of inhibition for PhnA driven by glucose [127]. In fact, when the organism grew on PnAc as the sole P source and pyruvate as the sole C source, phosphate was released. However, when glucose was used as the C source, no phosphate release was detected. Phosphate was also released when some disaccharides were used as C sources, indicating that when the organism is unable to use a given C source, it can metabolize the C contained in PnAc [127]. Glucose-driven repression was also shown when AEP was used as the P source but not when it was used as the sole N or N and P source [127].

The *Bulkhoderia cepacia* Pal6 phosphonopyruvate hydrolase, PalA, allows for the utilization of L-PnAla as the C, N, and P source by the environmental isolate [64]. In particular, cells were able to grow in phosphate-starved conditions and in the presence of PnAc as the sole P source thanks to C–P lyase activity. Instead, PalA activity was induced only when PnAla was present in the medium and regardless of the presence of phosphate [64]. These findings suggest that PalA may not belong to the Pho regulon, since its expression is phosphate-independent; instead, it is substrate-inducible, as in the case of *phnA* [64]. Similar results were obtained from studies on *Variovorax* sp. Pal2, where *palA* gene expression was induced only when the organism was grown on PnAla or PnAc and in a phosphate-independent manner [66].

### 7.3. LysR-Like Transcriptional Regulator Proteins Are Recurrent in Operons for C–P Hydrolysis

Since the ability (or inability) to degrade phosphonate is controlled at the stage of gene expression, rather than through the regulation of enzyme activity [8], and since there is clear evidence that phosphate-independent C–P bond cleavage occurs among both soil and marine microbes, alternative regulatory mechanisms should exist, not related to the PhoBR system. Evidence of this was found in studies on the C–P hydrolases neighboring regions, which led to the discovery of genes predicted to encode transcriptional regulators belonging to the LysR family.

For example, structural analysis of the *P. fluorescens* 23F *phnA* gene region has shown five more genes clustered with the hydrolase [53]. The products of three of them are putative transporters, while a gene found upstream *phnA* was named *phnR*, and its product showed high similarities to members of the LysR family of transcriptional regulators [53]. Studies on the regulation of PhnA activity have established the importance of the PhnR protein in gene expression. In fact, both PnAc and phosphonopropionate acted as inducers of *phnA* gene expression in clones possessing both PhnA and PhnR, while clones that possessed a non-functional PhnR were unable to grow on PnAc, suggesting that PhnR is necessary for the substrate-dependent induction of *phnA* [53].

The discovery and functional characterization of PhnR on *P. fluorescens* 23F paved the way for the construction of a biosensor capable of detecting low levels of PnAc as well as PAA and arsonoacetate with less sensitivity [137]. The PnAc “sensor region” involved the *phnR* gene with its promoter region, as well as part of the *phnA* gene with its promoter, and it was fused with the *gfp* gene [137]. Hence, with the presence of PnAc in the medium, this C–P compound acts as the co-inducer binding PhnR, which, in turn, induces *gfp* expression. With this biosensor, the lowest PnAc concentration detected was 500 nM, 100 times less than that of other available analytical methods [137].

The *S. meliloti phnWAY* cluster, as well as the *P. putida* NG2 *phnWX* cluster, both for AEP degradation, also include a divergently transcribed open reading frame, located upstream of the structural genes, which appears to encode a LysR-like transcriptional regulation protein [40,133]. Even in the absence of direct experimental evidence, the closeness to the *phnWAY* and *phnWX* genes led to the hypothesis that these putative transcriptional regulators could be involved, in some way, in the regulation of the pathways. On the other hand, the AEP utilization in *Pseudomonas putida* BIRD1 through the PhnWX pathway was shown to be under a complex dual regulation, which involves the CbrAB and NtrBC systems, as described above, and a LysR-like transcriptional regulator called AepR [126]. In fact, *phnWX* gene expression under C or N depletion occurred only when AEP was present as the C or N source, respectively, and when AepR was functionally active [126]. These findings revealed that AEP is likely to be the co-inducer which enables the substrate-inducible expression of *phnWX* through the LysR-like regulator.

An analysis of the neighboring region to the *Variovorax* sp. Pal2 *palA* gene led to the discovery of five other genes involved in PnAla uptake and degradation [23]. Among these, the *palB* gene encodes the enzyme involved in the first step of the pathway (Section 3), three other genes (*palC, palD,* and *palE*) encode putative transporters, and finally, *palR* encodes a protein highly similar to several transcriptional regulators belonging to the LysR family [66]. In this *Variovorax* strain, both a functional PalR and the presence of either PnAla or PnPyr are required to induce the gene operon related to PnAla degradation [66].

The transcriptional regulator proteins belonging to the LysR family are the most abundant and highly conserved among bacteria [138]. The LysR-dependent regulation involves the presence of a co-inducer which determines the protein affinity with the binding regions on the DNA and, thus, allows LysR to enhance or inhibit gene expression [138].

In conclusion, the three main C–P hydrolases, PhnX, PhnA, and PalA, are specific for different phosphonate compounds, which also act as the co-inducers for the activation of their respective catabolism. Such catabolism, in turn, would likely depend on LysR transcriptional regulators found near the structural genes.

## 8. The Microbial Degradation of Man-Made Phosphonates

Anthropogenic phosphonates pose a high environmental risk due to their extensive use in combination with a high water solubility [139]. As phosphonates began to be used in cleaning detergents, washing agents, and technical processes, their share of the DOP fraction started to increase significantly in free waters [140]. Today, several thousand tons of phosphonates are introduced into the environment every year. A substantial rise in phosphonate concentrations was also detected in the effluents of many European wastewater treatment plants (WWTPs), where they become increasingly problematic.

Among xenobiotic phosphonates, polyaminomethylenepolyphosphonates (polyphosphonates, PPns) and glyphosate (Figure 2) have the highest industrial importance, and their microbial degradation will thus be discussed in this section.

### 8.1. Polyphosphonates (PPns)

PPns typically contain more than two C–P bonds, and interest in them has grown due to their effective chelating properties. They form highly stable metal complexes with Fe, Mg, and Ca, especially at an alkaline pH [141], and thus proved to be useful additives in many industrial and household processes over the past 30 years. They are used in bulk quantities during textile and paper production, in household and industrial cleaners, and in the building materials industry; additionally, they are integral components of anticorrosives and fire retardants, are essential additives during membrane filtration, and are approved for the treatment of various bone diseases. The quantitatively most important individual compounds of this group are 2-phosphonobutane-1,2,4-tricarboxylic acid (PBTC), 1-hydroxyethane 1,1-diphosphonic acid (HEDP), nitrilotris(methylene phosphonic acid) (NTMP), ethylenediamine tetra(methylene phosphonic acid) (EDTMP), and diethylenetriamine penta(methylene phosphonic acid) (DTPMP) (Figure 2) [142]. They were shown to inhibit algal growth and oyster shell formation at relatively low concentrations, presumably due to their complexing properties, and to effect varied fish populations [13,143].

In the early 1990s, the Europe-wide consumption of polyphosphonates (excluding PBTC) was estimated to be 11,820 t/a. [141]. These numbers rose dramatically until 2012, when the European phosphonate consumption alone reached 49,000 t and the world-wide consumption reached 94,000 t, which equals a 70% increase between 1998 and 2012 [13]. Notably, Europe is the continent with the highest phosphonate consumption, and approximately 85% of the PPns in WWTPs stem from domestic use [144].

The overall microbial degradability of PPns was found to be low, and degradation rates typically range from 0 to 40%, presumably due to their slow uptake [143]. Today, various microorganisms are known to degrade biogenic and xenobiotic phosphonates, including bacteria, fungi [145], and green algae [146]. Among all studied microbes, *Arthrobacter* sp. GLP-1 was the only one capable of degrading HEDP, NTMP, EDTMP, and DTPMP. Most other studied organisms were found to only degrade one group of phosphonates, such as *Anabaena variabilis* (degrades DTPMP to *N*-acetyl-aminomethylphosphonate (AcAMP)) or *Spirulina platensis*, which effectively degrades 50% of HDTMP to AMPA (Figure 2).

Studies on PPn degradation further show a general trend towards the faster microbial degradation of nitrogen-containing PPns [13], and PPn degradation was shown to occur preferentially under phosphate-free conditions. There do not seem to be significant differences in the degree of phosphonate degradation between sludge samples which should be adapted to phosphonates and natural environmental samples [107]. Thus, PPn-degrading microorganisms seem to be ubiquitous but not necessarily abundant.

The biodegradation of PPns is the subject of much active research [147]. While the microbial breakdown of these compounds is generally a slow process, their contribution to processes such as eutrophication should not be underestimated [13]. Their microbial degradation occurs in parallel to abiotic processes such as catalytic oxidation and photolytic decomposition, especially in the presence of iron salts and at an acidic pH [78,141]. In experimental set-ups, these abiotic processes were shown to be the predominant route for PPn degradation in aqueous systems, with orthophosphate (93%) and AMPA (7%) being the major products. As a consequence, even microorganisms lacking enzymes for phosphonate degradation can indirectly use PPns as a phosphorus source [148].

### 8.2. Glyphosate

The single most important anthropogenic phosphonate is certainly the herbicide glyphosate (Figure 2). It was commercialized in 1974, and ever since, glyphosate production and use increased. Today, glyphosate sales account for more than 25% of the global herbicide market [25]. It interferes with the enzyme 5-enolpyruvylshikimate-3-phosphate synthase (EPSPS) of the shikimic acid pathway and thus inhibits the biosynthesis of aromatic amino acids and plant hormones [149].

Despite the controversies about the potential environmental and health issues related to extensive glyphosate use, the global production increased from 620,000 t/a in 2008 [150] to 825,000 t/a in 2014 [151]. Between 1995 and 2014, the use of glyphosate in agricultural and domestic herbicides has risen by over 12-fold, and between 2004 and 2014, 6.1 billion kg of glyphosate were used, which equals 72% of the entire used amount [152].

As a result of its extensive use, glyphosate concentrations far above the legal limits were reported in many ecosystems, and it is known to widely occur in agricultural soils, surface water sediments, and shallow groundwaters around riverbanks [153]. Rare examples even report its presence in deep groundwater. Studies show that almost 52% of the sprayed glyphosate reaches surface water bodies, mainly by deposition, and it is estimated that approximately 1% of the global cropland is polluted by glyphosate [154].

Even though microbial organisms generally rely on EPSP for the synthesis of aromatic amino acids, several resistance mechanisms towards glyphosate developed, which include the synthesis of EPSP that is insensitive towards glyphosate [77], the synthesis of an increased amount of EPSP, and the downregulation of the major glyphosate transporters [42,77]. The phosphorylation [155] or acetylation [156,157] of glyphosate are equally known detoxification strategies.

As a result, glyphosate is more readily used by microbes than PPns, and microbial degradation is the only known degradation process of glyphosate in soil [158]. Among the studied microbial species, most use glyphosate as the phosphorus source, with a few exceptions using it as a nitrogen [159,160] or carbon [161] source. Today, most known glyphosate degraders are either aerobic or facultatively anaerobic bacteria [162], with *Pseudomonas* spp. being the biggest bacterial group of glyphosate degraders [149,163]. The ability to degrade glyphosate is also known in the fungal genera *Penicillium*, *Aspergillus*, *Fusarium*, *Scopulariopsis*, and *Trichoderma* [145,164,165,166], with degradation efficiencies ranging from 40 to 100% [25]. Glyphosate degradation was also observed in plants [167] and animal cells [168] but is most likely attributed to associated microorganisms there.

In general, glyphosate degradation occurs via two different pathways, both of which involve C–P lyase and either start with C–P bond cleavage (sarcosine pathway) or C–N bond cleavage (AMPA pathway) (Figure 8) [98]. However, it was noticed that there seems to be a glyphosate-specific variant of C–P lyase present in some bacterial strains that functions differently with respect to the well-studied version from *E. coli* [86,87]. While glyphosate is not a substrate for C–P lyase from *E. coli* and *S. stutzeri*, it is accepted by *Ochrobactrum antrhopi*, *S. meliloti, Agrobacterium radiobacter*, *Burkhoderia pseudomallei*, and *Nostoc* sp. [77], as well as by C–P lyases from naturally isolated *Klebsiella* and *Pantoea* strains [169]. Bacterial strains with glyphosate-specific C–P lyase are typically highly efficient in their degradation under laboratory conditions. However, this often changes in natural ecosystems due to several factors, such as the fact that C–P lyase is expressed only under phosphate starvation conditions [149].

The dominant pathway for glyphosate usage is the AMPA pathway [3], where first, a C–N bond is broken by the flavine-dependent glyphosate oxidoreductase to yield AMPA and glyoxylate. The latter can be readily used as a carbon source by many microbial strains [170]. The resulting AMPA is, however, microbially very stable. Most microorganisms release the resulting AMPA to the environment, suggesting that genes for glyphosate detoxification and for phosphonate use as a P-source developed independently.

AMPA is also the main microbial degradation product of NTMP, EDTMP, and DTPMP in soil [171]. Its environmental relevance is controversial, and it was shown to cause DNA and chromosomal damage in fish [172]. Each year, 70 t of AMPA is released to surface waters from soil, with 63 t derived from glyphosate and 7 t derived from other aminophosphonates. Generally, AMPA degradation is a microbiological process, which increases with increasing humidity and temperature due to increased microbial activity and growth. Its soil half-life heavily depends on the exact composition of the microbial community and can range from 10 to 100 days.

There are, however, a few microbial strains [173,174,175] that effectively degrade AMPA, either by hydrolytic enzymes (phosphonatase) after transamination (a corresponding glyphosate-specific aminotransferase was identified in *O. anthropi* GPK 3 [176]) to give phosphate and formaldehyde [87] or by C–P lyase to give methylamine and phosphate [160]. Of these, AMPA degradation by C–P lyase is the most common strategy but only becomes possible after acetylation by PhnO (see Section 5.3) [177]. The formed AcAMPA can enter the classical C–P lyase degradation pathway and finally lead to a release of phosphate and *N-*methylacetamide. The latter can be further degraded to CO_2_ in several microorganisms [160].

The second pathway for glyphosate degradation is known as the sarcosine pathway after its major intermediate. In this case, glyphosate directly undergoes C–P bond cleavage by C–P lyase to give sarcosine and phosphate [174]. While the sarcosine pathway is favorable from a thermodynamic point of view (producing both N and P as nutrients), after very short culture times (typically within 10 days), most microbial cultures favor the AMPA pathway. This observation was speculated to prevent ammonia overflow in cells. However, there are certainly also other factors influencing the shift between both known major glyphosate degradation pathways such as the availability of phosphate [77,178]. As a consequence, the sarcosine pathway has so far only been observed in experimental set-ups. The ability to degrade glyphosate to sarcosine was, however, equally detected in bacteria from glyphosate-polluted soil and in strains that were never exposed to glyphosate before. This suggests the existence of natural compounds of a similar structure that might serve as substrates for the involved glyphosate-specific C–P lyase [176]. The resulting sarcosine can be further metabolized to glycine and formaldehyde under the action of sarcosine oxidase and thus finally yield CO_2_ and NH_3_ [158].

Apart from these two major microbial degradation pathways, other enzymes are also known to be involved in glyphosate degradation such as glycine oxidase [177].

## 9. Concluding Remarks

The microbial degradation of phosphonates is a very active area of research, given the wide-ranging involvement of phosphonates in biology, ecology, and geochemistry. Studies in the field keep revealing a remarkable diversity of catalytic tools and biochemical pathways that microorganisms employ to catabolize phosphonates [1,8,23,179]. This, in turn, provides a better understanding of the role of phosphonates in the global biogeochemical P-cycling and may be helpful in view of bioremediation efforts concerning the anthropogenic phosphonate pollutants.

As outlined in this review, a large number of studies in the last few decades have led to exciting discoveries and to a much better understanding of the topic. However, many challenges remain ahead, a few of which are outlined below.

For example, future work investigating the regulation of C–P lyase and organisms harboring variant C–P lyase operons including archaeal organisms would provide an in-depth understanding of the uses of the enzyme system in nature.

Additionally, while the biochemistry of phosphonate degradation is increasingly well understood, the biogeochemical role of these pathways is still comparatively poorly characterized. While traditionally viewed as a method for obtaining P by P-starved microorganisms, there is an increasing amount of culture, metagenomic, and transcriptomic data showing that phosphonates can be consumed under abundant phosphate levels as a source of N or C. Further study of phosphonate turnover may reveal whether providing phosphorus is the major environmental function of phosphonate catabolism pathways or if this catabolism is largely driven by the need to obtain C or N or by other factors entirely.

Furthermore, while the scope of this review was focused on phosphonate catabolism, characterizing the reasons for synthesizing these molecules will also be needed to complete our understanding of their environmental metabolism. Recent work has shown that bacteria can produce specific phosphonate products as antimicrobials and pathogenic agents (e.g., [180,181]), but the function(s) of common phosphonates like AEP and MePn remain poorly defined. There is some evidence to suggest that the production of P compounds including phosphonates may facilitate interactions between closely associated microbes [116] and that they may also protect cells from grazing and lysis [182]. Attempting to empirically demonstrate these functions will be essential for understanding how and when these molecules become available for catabolism.

Finally, while there is much left to discover about phosphonate cycling in prokaryotes, phosphonates remain even more cryptic in eukaryotes, even though these molecules were originally found in eukaryotic cells. The recent progress that has been made in phosphonate biochemistry has significantly improved our understanding of how these molecules can be catabolized, but much remains to be learned about their synthesis and function in cells and environments.

## Figures and Tables

**Figure 1 molecules-28-06863-f001:**
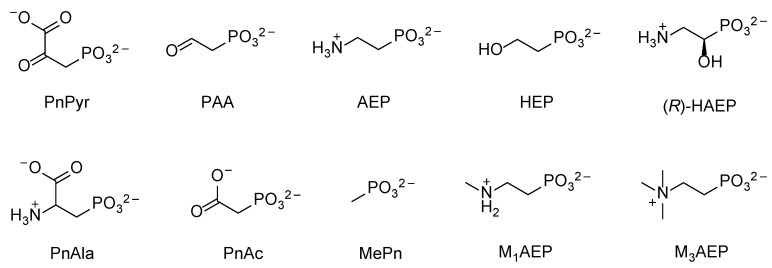
The most common natural phosphonates. Phosphonopyruvate (PnPyr) is the precursor of nearly all known natural phosphonates. It is formed from the glycolytic intermediate phosphoenolpyruvate via the reaction of phosphoenolpyruvate mutase [1] coupled with one of four known irreversible processes (in particular, the decarboxylation of PnPyr to yield phosphonoacetaldehyde—PAA), which serves to thermodynamically ‘drive’ the formation of the stable C–P bond.

**Figure 2 molecules-28-06863-f002:**
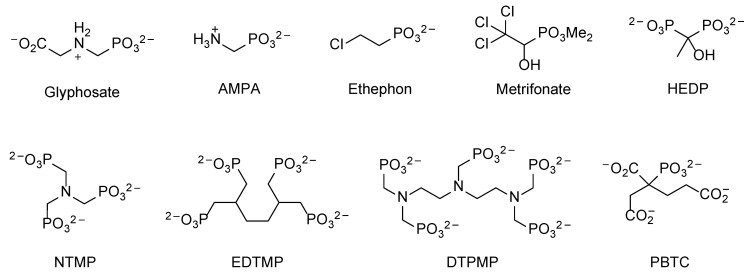
Some man-made phosphonates that may occur in the environment. Glyphosate is the most widely used herbicide. AMPA is a long-lasting breakdown product of glyphosate. Ethephon is a plant growth regulator (ultimately converted into ethylene by plants) that is used to induce the ripening of various crops. Metrifonate is used as an insecticide in restricted applications. 1-Hydroxyethane 1,1-diphosphonic acid (HEDP), nitrilotris(methylene phosphonic acid) (NTMP), ethylenediamine tetra(methylene phosphonic acid) (EDTMP), and diethylenetriamine penta(methylene phosphonic acid) (DTPMP) are chelating agents used in water treatment as antiscalants and anti-corrosion agents [13].

**Figure 3 molecules-28-06863-f003:**
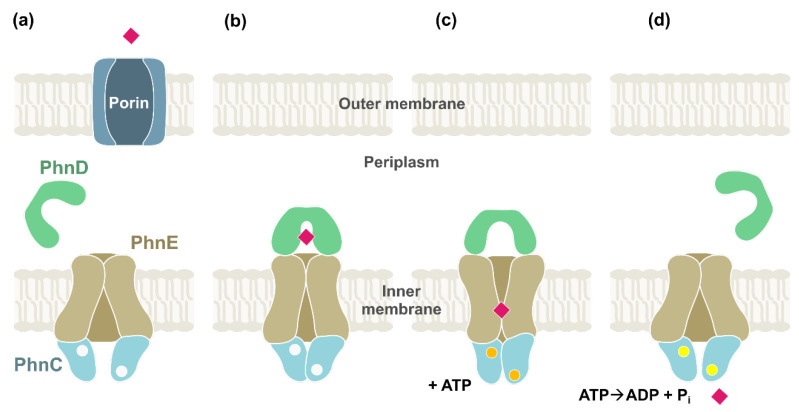
In *E. coli* and many other Gram-negative bacteria, phosphonates such as 2-aminoethylphosphonate (AEP, also known as ciliatine; see Figure 1) can be transferred from the periplasm to the cell interior by an ABC-type transporter called PhnCDE. (**a**) When the ligand (symbolized by the red diamond) is not present in the periplasm, the substrate-binding protein (PhnD) is not associated with the transporter. Presumably, phosphonates enter the periplasm through the same porins exploited by other phosphate compounds—for example, through PhoE. PhoE is a porin which allows for the diffusion of several phosphorus-containing molecules, is under the control of the Pho regulon (see Section 7), and is expressed when the levels of environmental phosphorus are low [28,31,32]. (**b**) Upon the binding of the phosphonate, PhnD interacts with the transmembrane complex and the nucleotide binding domains of PhnC move towards each other. (**c**) ATP (symbolized by the orange circle) binds to PhnC and the channel moves to an outward conformation, allowing the substrate to enter. (**d**) ATP is hydrolyzed (circle changes from orange to yellow) and the channel adopts an inwards conformation, allowing the phosphonate to diffuse into the cytoplasm.

**Figure 4 molecules-28-06863-f004:**
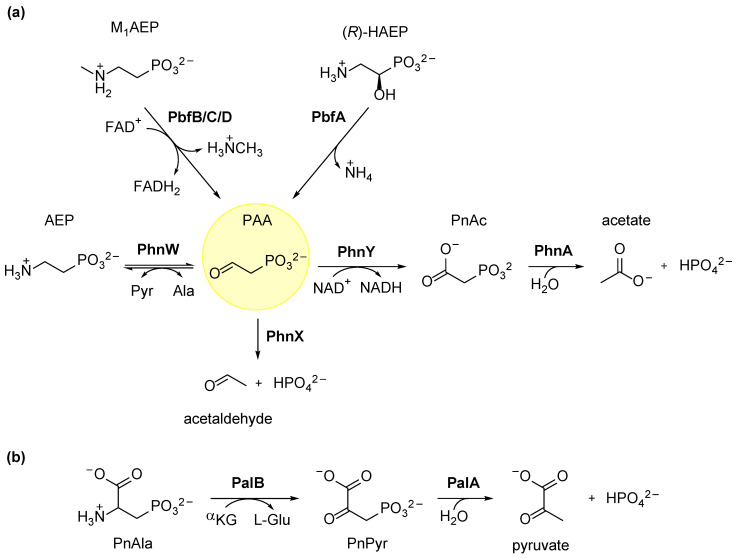
Schematic representation of the hydrolytic pathways. (**a**) The PhnWX and PhnWAY pathways share the common intermediate PAA, which is produced from AEP by the enzymatic activity of the transaminase PhnW. PAA can also be derived from (*R*)-1-hydroxy-2-aminoethyl phosphonate-(*R)*-HAEP-by the PLP-dependent lyase PbfA [45] or obtained from the oxidation of *N*-methyl AEP (M_1_AEP), operated by some ancillary FAD-dependent enzyme (PbfB, PbfC, or PbfD) [46]. PAA is then further processed by either the hydrolase PhnX (PhnWX pathway) or the dehydrogenase PhnY (PhnWAY pathway). (**b**) The PalAB pathway allows for PnAla mineralization.

**Figure 5 molecules-28-06863-f005:**
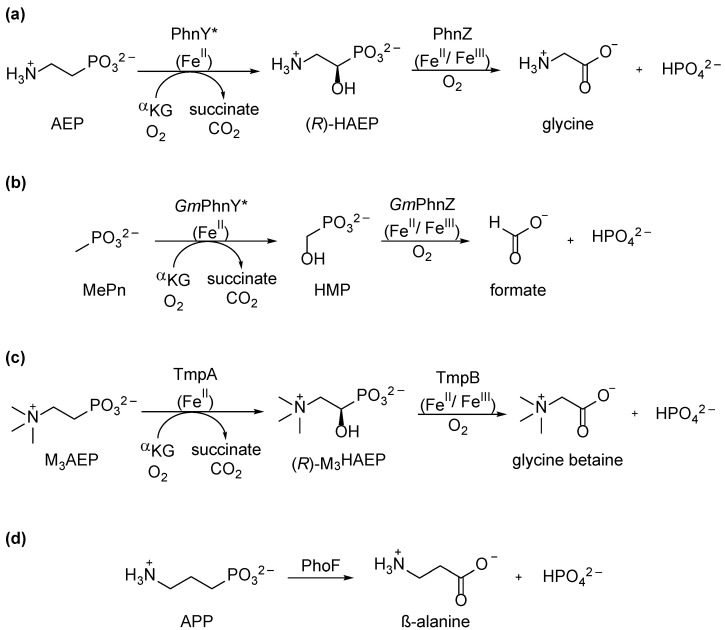
Oxidative degradation pathways for phosphonates catalyzed by (**a**) PhnY*/PhnZ, substrate: AEP, (**b**) *Gm*PhnY*/*Gm*PhnZ, substrate: MePn, (**c**) TmpA/TmpB, substrate: M_3_AEP, and (**d**) PhoF, substrates: AEP and 3-aminopropyl phosphonate (APP).

**Figure 6 molecules-28-06863-f006:**
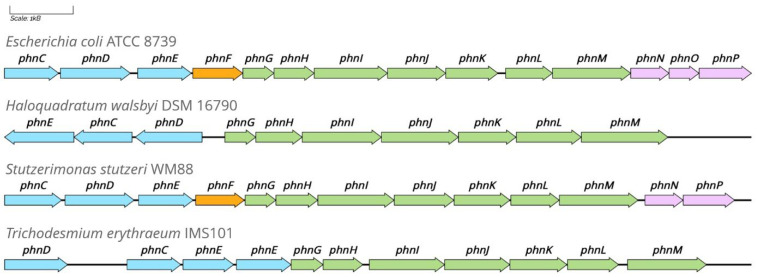
Operon structures of the C–P lyase complex in *E. coli* ATCC 8739 (Genbank accession CP033020.1) *Haloquadratum walsbyi* DSM 16790 (Genbank accession AM180088.1) *Stutzerimonas stutzeri* (Genbank accession AY505177.1) and *Trichodesmeium erythraeum* IMS 101 (Genbank accession CP000393.1). Subunits are colored by function. Genes encoding transport proteins are shown in blue (*phnCDE*), the regulatory gene phnF is shown in orange, the ‘core’ C–P lyase genes are shown in green (*phnGHIJKLM*), and the accessory genes (*phnNOP*) are shown in pink [90]. Note the variation in the presence of accessory genes as well as the *phnF* regulator. Gene lengths are proportionate, with the scale bar at the top equal to 1 kb.

**Figure 7 molecules-28-06863-f007:**
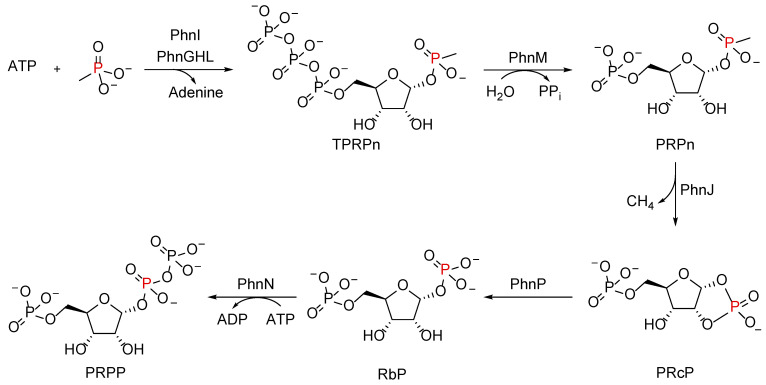
Mechanism and products of the reaction catalyzed by PhnGHIJKLMOP of the C–P lyase pathway, with MePn as an example substrate. The phosphonate-derived phosphorus atom is highlighted in red. ATP: adenosine triphosphate. RTP: D-ribose-5-triphosphate. RPnTP: α-D-ribose-1-methylphosphonate-5-triphosphate. PRPn: α-D-ribose-1-methylphosphonate-5-phosphate. PRcP: α-D-ribose-1,2-cyclic-phosphate-5-phosphate. RbP: Ribose-1,5-bisphosphate. PRPP: phosphoribosyl pyrophosphate (α-D-ribose-1-diphosphate-5-phosphate).

**Figure 8 molecules-28-06863-f008:**
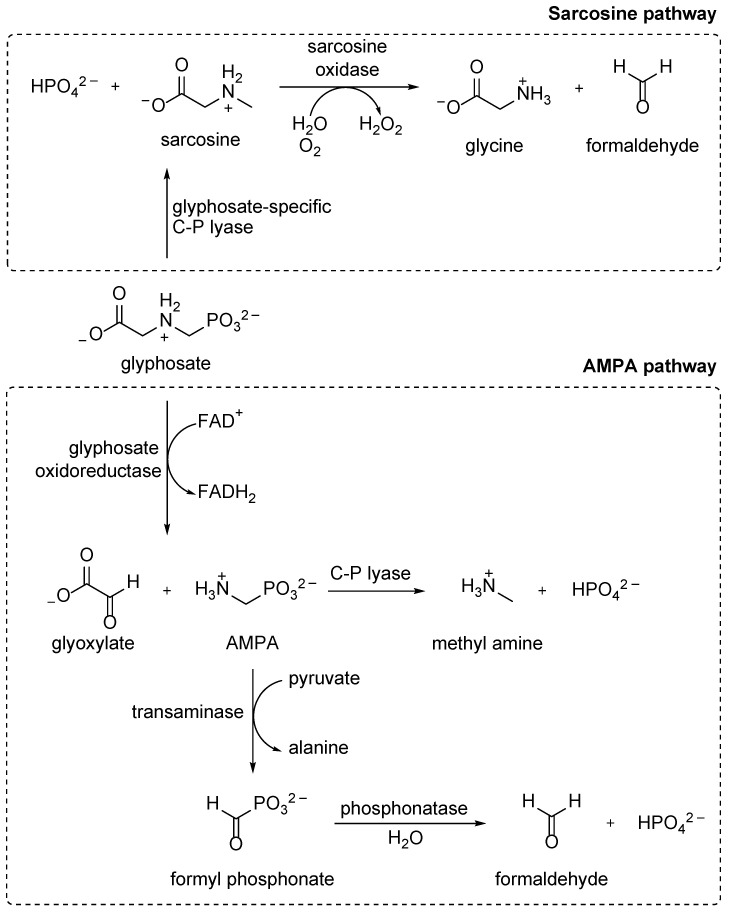
Known microbial pathways for glyphosate degradation.

**Table 1 molecules-28-06863-t001:** Abundance of phosphonate catabolism genes in surface waters of metagenomic databases, assessed in different studies. Values for refs. [3,8] are taken directly from the text, while those for refs. [36,39] are approximated from the respective figures. All values are the average percentage of expected genomes per sample containing the gene, except for Murphy et al. 2021 [39], who presented the median percentage of expected genomes per sample. nd: not determined.

Gene	Gene Abundance (Percentage of Expected Genomes)			
	Martinez et al. [8]	Villarreal-Chiu et al. [3]	Murphy et al. [39]	Lockwood et al. [36]
*phnX*	1%	1.8%	<0.1%	~2%
*phnA*	9%	11.2%	<10%	~10%
*palA*	nd	0.1%	nd	<5%
*phnZ*	7%	9.4%	<1%	~20%
*phnJ*	8%	5.1%	<1%	>10%

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
