# Peer review of "The Microbial Degradation of Natural and Anthropogenic Phosphonates"

_molecules, 2023, doi:10.3390/molecules28196863_

Round 1

Reviewer 1 Report

1. Authors should discuss the timeframe that was covered in the current review along with a brief overview of any prior reviews that were pertinent to the subject and what was not addressed there and was added here.

2. The majority of the cited papers are older than ten years, and there aren't many recent references at all. It is preferable to include recent references within the last five years

3. Figure 3's image quality is poor; it is blurry. Replace it with a high-quality image.

No objection

Author Response

Referee 1

  1. Authors should discuss the timeframe that was covered in the current review along with a brief overview of any prior reviews that were pertinent to the subject and what was not addressed there and was added here.

Reply: The MDPI review article description says that “Reviews offer a comprehensive analysis of the existing literature within a field of study”. So there is not a specific timeframe to be covered. However, we now cite, at the end of the discussion, the previous reviews pertaining to the topic and mention briefly some of the most recent material covered by our paper.

  1. The majority of the cited papers are older than ten years, and there aren't many recent references at all. It is preferable to include recent references within the last five year.

Reply: In the original submission, 51 references out of 178 (28.7%) were to papers published in the last five years. We believe such a fraction was fair for a review that, as stated above, aims at being a comprehensive perspective on the topic, rather than just a collection of ‘latest news’. Citations from older works served (and still serve) to provide background information that is essential to understand the state of the art. At any rate, the revised paper cites (and when appropriate discusses) a few additional recent papers, including an article from our own labs that was just accepted for publication. Accordingly, the number of post-2017 references has increased to 56 out of 182 (30.8%) in the revised manuscript. On the other hand, we felt it inappropriate to cite papers that, albeit recent, were exceedingly marginal to the subject of this review.  

  1. Figure 3's image quality is poor; it is blurry. Replace it with a high-quality image.

Reply: We have modified the figure to meet the referee’s comment.

Reviewer 2 Report

This is a high-quality and well-organized manuscript that broadly covers the microbial degradation pathways and mechanism of natural phosphates and anthropogenic compounds. From bacterial transporters, catabolism pathways, and C-P lyase complex to the genomic analysis, the authors throughly described from the background to the current progress on the gene-to-protein levels of phosphate catabolism, including a variety of microbes from marine system to soil system and the regulatory pathways.

Only one suggestions is that figure 3 is a little fuzzy.

Author Response

Referee 2

This is a high-quality and well-organized manuscript that broadly covers the microbial degradation pathways and mechanism of natural phosphates and anthropogenic compounds. From bacterial transporters, catabolism pathways, and C-P lyase complex to the genomic analysis, the authors throughly described from the background to the current progress on the gene-to-protein levels of phosphate catabolism, including a variety of microbes from marine system to soil system and the regulatory pathways.

Reply: We thank the referee for her/his appreciation.

Only one suggestions is that figure 3 is a little fuzzy.

Reply: We have modified Figure 3, also including a higher resolution version in the text.

Reviewer 3 Report

This review by Ruffalo and others is devoted to the analysis of natural and anthropogenic phosphonates and their microbiological degradation. Phosphonates are a class of organophosphorus compounds characterized by a chemically stable carbon–phosphorus bond. Only microorganisms are capable of biodegradation of phosphonates in several ways. The destruction of the inactivated C–P bond by means of is of fundamental importance, and understanding this process is the main problem of biochemistry and physiology of microorganisms. The review mainly deals with modern articles, where data on microorganisms decomposing phosphonates are analyzed. The review is written in a clear and intelligible language. The only edits on the review relate to the list of references. Check all links to the subject of registration according to the rules of the MDPI journal.

Author Response

Referee 3

This review by Ruffalo and others is devoted to the analysis of natural and anthropogenic phosphonates and their microbiological degradation. Phosphonates are a class of organophosphorus compounds characterized by a chemically stable carbon–phosphorus bond. Only microorganisms are capable of biodegradation of phosphonates in several ways. The destruction of the inactivated C–P bond by means of is of fundamental importance, and understanding this process is the main problem of biochemistry and physiology of microorganisms. The review mainly deals with modern articles, where data on microorganisms decomposing phosphonates are analyzed. The review is written in a clear and intelligible language.

Reply: We thank the referee for her/his positive comments.

The only edits on the review relate to the list of references. Check all links to the subject of registration according to the rules of the MDPI journal.

Reply: This comment is a bit cryptic. We see no obvious discrepancy between the style of the references in our submission and the style of the references in the ‘Molecules’ article template from MDPI. As a matter of fact, we used a reference management software (Mendeley) that automatically implements the journal style to the references list, so the possibilities of errors are limited. But if the referee could specifically pinpoint the problems that escaped our (and Mendeley's) attention, we’ll be glad to correct our text.